# Live Cell Poration by Au Nanostars to Probe Intracellular Molecular Composition with SERS

**DOI:** 10.3390/nano11102588

**Published:** 2021-09-30

**Authors:** Evelina I. Nikelshparg, Ekaterina S. Prikhozhdenko, Roman A. Verkhovskii, Vsevolod S. Atkin, Vitaly A. Khanadeev, Boris N. Khlebtsov, Daniil N. Bratashov

**Affiliations:** 1Department of Biophysics, Biological Faculty, Moscow State University, 1-12 Leninskie Gory, 119991 Moscow, Russia; 2Science Medical Center, Saratov State University, 83 Astrakhanskaya, 410012 Saratov, Russia; prikhozhdenkoes@sgu.ru (E.S.P.); r.a.verhovskiy@sgu.ru (R.A.V.); atkin.vsevolod@gmail.com (V.S.A.); khlebtsov_b@ibppm.ru (B.N.K.); 3Institute of Biochemistry and Physiology of Plants and Microorganisms, Russian Academy of Sciences, 13 Prospekt Entuziastov, 410049 Saratov, Russia; khanadeev_v@ibppm.ru; 4Veterinary Medicine and Biotechnology Faculty, Saratov State Agrarian University, 1 Teatralnaya Square, 410012 Saratov, Russia; 5Moscow Institute of Physics and Technology, 9 Institutskiy per., 141701 Dolgoprudny, Russia

**Keywords:** SERS, Au nanostars, cell membrane

## Abstract

A new type of flat substrate has been used to visualize structures inside living cells by surface-enhanced Raman scattering (SERS) and to study biochemical processes within cells. The SERS substrate is formed by stabilized aggregates of gold nanostars on a glass microscope slide coated with a layer of poly (4-vinyl pyridine) polymer. This type of SERS substrate provides good cell adhesion and viability. Au nanostars’ long tips can penetrate the cell membrane, allowing it to receive the SERS signal from biomolecules inside a living cell. The proposed nanostructured surfaces were tested to study, label-free, the distribution of various biomolecules in cell compartments.

## 1. Introduction

Analysis of live cell biochemistry attracts much interest nowadays. Surface-enhanced Raman scattering (SERS) is a valuable tool for such research as it is minimally invasive and allows online tracking of chemical composition and reactions in cell growth media [1,2,3,4,5,6,7,8]. The main problem is to ensure contact of the SERS probe with internal cellular structures, organelles, and the nucleus. Biocompatible SERS-active nanoparticles can be internalized inside the cell to study various molecules within a living cell [9,10,11,12], changes in protein conformation during mitosis [13], and spatio-temporal changes in cells during differentiation [14]. SERS active nanoparticles can be navigated inside the cell with magnetic [15], optical [16], or other kinds of tweezers. Combined labeled and non-labeled gold nanoparticles have been used to study the localization and molecular compositions of the plasma membranes and nuclei in cells [17]. Despite the variety of existing colloidal SERS probes, there are many restrictions on their use, including osmotic shock, the presence of toxic byproducts remaining after chemical synthesis, uncontrolled aggregation, and the need for an internalization procedure [4]. Currently, there is a trend towards using nanostructured surfaces, which are more stable and convenient for transportation and storage.

There are two possible mechanisms for enhancing the Raman signal of intracellular molecules on the SERS sensor surface. The first is a long-range enhancement when the SERS platform is outside the cell wall and examines molecules inside the cell. Aggregates of silver nanoparticles [18] and composite silica-silver nanostructures made it possible to amplify the Raman signal at large distances from submembrane molecules inside intact mitochondria or erythrocytes [19]. Nanostructured silver surfaces were used for label-free studies of conformational changes in cytochrome C heme in functional mitochondria adsorbed on its surface [20,21]. The SERS platform, consisting of graphene-coated gold nanopyramids, was used to study the intracellular composition, which made it possible to distinguish the p53 knockout cancer cell line [22].

There are many works on the synthesis of SERS probes in the form of gold nanostars. Many authors obtained Au nanostars through the surfactant-included procedure with the mixture of poly(vinyl pyrrolidone) and *N*,*N*-dimethylformamide (PVP/DMF) [23,24,25,26]. The new way of synthesis of such nanostructures is a surfactant-less approach with a quick reaction between AgNO_3_ and HCl followed by Au^+^ reduction to metallic gold using ascorbic acid [27,28,29,30,31]. Such star-shaped gold nanoparticles are thermodynamically unstable and susceptible to processes similar to Ostwald ripening, so they are usually placed on a polymer layer to provide some stabilization. Among the suitable polymers, the poly(4-vinyl pyridine) (PVPyr) has shown promising results up to now [32]. Unlike gold nanostars synthesized in a PVP/DMF mixture, stars without surfactant have fewer long sharp needles [33].

The star-shaped shape of these nanostructures provides good localization of the electromagnetic field enhancement near the tips [31,33,34,35] and, thus, effective SERS detection of chemisorbed (such as 1-naphtalenethiol [23,24], 4-mercaptobenzoic acid [30]) and physisorbed (such as crystal violet [30], Alexa Fluor 750 C5-maleimide [24]) substances. For proof-of-concept experiments of cell wall penetration, the star-based SERS tags with different Raman reporters were developed [25,27,31,36]. Nanostars adsorbed on polymers were used as an optoporating systems allowing to introduce vectors encoding fluorescent proteins directly into the living cells [37]. The ability of nanostars to internalize into cells has been used to generate an enhanced two-photon photoluminescence signal to enable cell tracking [38]. Another interesting Au nanostars application is developing planar SERS substrates using different polymer layers on either a glass or silicon wafer [28,30], with which good sensitivity has already been demonstrated with simple, pure chemical analytes such as 4-aminothiophenol.

We first utilized the penetrating ability of gold nanostars for vital intracellular SERS imaging. Our approach is based on creating plasmonic nanostructured surfaces of Au nanostars immobilized on a poly(4-vinyl pyridine) polymer layer on a glass substrate. The proposed nanostructures provide substantial enhancement of the Raman signal from DNA, RNA, lipids, proteins, and hemes inside living cells, thus allowing one to map molecular distribution in different cellular compartments.

## 2. Materials and Methods

### 2.1. Materials

Poly(4-vinyl pyridine) (PVPyr, 160 kDa), 4-Mercaptobenzoic acid (4-MBA), hydrogen tetrachloroaurate trihydrate (HAuCl_4_·3H_2_O), silver nitrate (AgNO_3_; >99%), HCl, and ascorbic acid were purchased from Sigma-Aldrich (Darmstadt, Germany). Fetal bovine serum (FBS) and Dulbecco’s modified Eagle medium (DMEM) were purchased from Life Technologies (Waltham, MA, USA). Calcein AM cell dye was purchased from Invitrogen (Waltham, MA, USA). Dulbecco’s phosphate-buffered saline (DPBS) was purchased from (BioloT, Saint Petersburg, Russia). Gold nanostars (Au nanostars) were obtained by the protocol described earlier [29] using 150 μL of 10 nm Au seeds from Sigma-Aldrich. All chemicals were used as received without further purification. Deionized water (specific resistivity higher than 18.2 MΩ·cm) from Milli-Q Direct 8 (Millipore, Merck KGaA, Darmstadt, Germany) water purification system was used to prepare all solutions.

### 2.2. Preparing of SERS Substrate

Prior to preparing SERS substrates with gold nanostars, cover glass slides (24 × 24 mm, Leica, Wetzlar, Germany) were subjected to cleansing using piranha solution (H_2_SO_4_:H_2_O_2_ (30% solution) in 3:1 ratio). Then, clean cover glass slides were washed and immersed in PVPyr solution (2 mg/mL) for 15 min. After subsequent washing with water, glass-PVPyr slides were put into Petri dish and covered overnight with a suspension of freshly synthesized Au nanostars. The next day, the prepared SERS substrates were washed from the residual unbound Au nanostars.

### 2.3. Characterization

SERS signals of HeLa cells were measured by the Renishaw inVia Raman microscope (New Mills, Gloucestershire, UK) using 50×/0.5 n.a. objective lens, laser wavelength 785 nm, and laser power 3 mW (100% = 30 mW). SERS measurements of 4-MBA (10${}^{-5}$ M solution in ethanol) were performed with the same setup using different laser power (0.3 mW) and time per single spectrum (1 s). SEM images of samples were taken with a Tescan MIRA II LMU (Tescan, Brno, Czech Republic) at the 30 kV accelerating voltage. Absorbance spectra of Au nanostars were measured by a Synergy H1 Multi-Mode plate reader (BioTek Instruments, Inc., Winooski, VT, USA). The spectra were obtained in the 400–999 nm spectral range with a 1 nm step. Cell viability and adhesion properties were observed using an Olympus IX73 inverted microscope (Tokyo, Japan). AFM images of the substrate, polymer layer, and surface covered with nanostars aggregates were obtained with an NTEGRA Spectra microscope (NT-MDT Spectrum Instruments, Zelenograd, Moscow, Russia) in tapping mode. NSG10 probes from NTMDT-SI with a typical resonance frequency around 220 kHz and tip curvature below 10 nm were used for image acquisition. All subsequent image processing was carried out with Gwyddion software [39]. The layer thicknesses were obtained by scratching the soft layer and measuring the height difference at the scratch area. The image background was subtracted, and peaks on the height distribution function were used to measure the thickness as described before [40]. Confocal laser scanning microscopy (CLSM) images were obtained using Leica TCS SP8 X (Leica, Wetzlar, Germany).

### 2.4. Cell Culture and Targeting

The human cervix carcinoma cell line (HeLa) was kindly donated by Shemyakin-Ovchinnikov Institute of Bioorganic Chemistry of RAS and cultured in DMEM supplemented with 10% FBS and 1% penicillin-streptomycin antibiotic antifungal cocktail under a humidified atmosphere with 5% CO_2_ at 37 °C. HeLa cells were seeded onto a glass coverslip (a), or cover slip covered with PVPyr (b), or SERS substrate (c) placed into a 35 mm non-modified plastic Petri dish (100,000 cells per dish) and incubated overnight in 2 mL supplemented DMEM culture medium. Substrates for cell growth (a–c) were previously exposed to UV for 20 min on both sides for sterilization.

The murine melanoma cell line (B16-F10) was provided by the Department of Cell Engineering, Education and Research Institute of Nanostructures and Biosystems, Saratov State University, Saratov, Russia.

### 2.5. Living Cell Imaging

Using prior measurements, cells were rinsed two times with 37 °C DPBS, and 2 mL DPBS was added. To prevent the appearance of peaks from the plastic bottom of the Petri dish, a sterile silicon wafer was placed under the substrate with cells in a Petri dish. SERS imaging of living cells was performed in a Streamline HR regime of Renishaw inVia Raman microscope with 1 μm steps using 50×/0.5 n.a. objective lens, laser wavelength 785 nm, laser power 3 mW. At each scanning point, the signal was collected for 1 s. Spectra were analyzed with open source software Pyraman [41].

For CLSM imaging, Calcein AM was added 1:1000 to culture media to stain the living cells and left for 30 min. All imaging was conducted in glass-bottom (0.17 mm thick) Petri dishes.

### 2.6. Processing of Raman Imaging Data

Regions surrounding the aggregate of Au nanostars inside the HeLa live cell were chosen for SERS imaging. Images were analyzed with Pyraman using the following algorithm [42]. Baseline was subtracted in each spectrum using the same recipe chosen after analyzing 10–15 spectra from the image. Root-mean-square (RMS) value was calculated for Raman intensities in specific frequency ranges. The further processing of Raman images is described in the Results and Discussion section.

## 3. Results and Discussion

### 3.1. Nanostructure Characterization

The structure of the SERS substrate is schematically shown in Figure 1A. The overview SEM image (Figure 1B) clearly shows aggregates of nanostars with sharp needles. The measured diameter of the nanostar core was 40.0±1.2 nm; the sharp needles had a length of 36.9±5.8 nm. More discussion on nanostar size statistics and its influence on physical properties for stars obtained with the similar synthesis can be found in [43]. The sensor consists of a glass substrate, covered by the thin layer of PVPyr with the deposited Au nanostars. As a result, a developed surface is formed with chaotic pile-ups of stars on the polymer layer from small clusters to large aggregates 0.5–0.6 μm in height. Surface coverage varied between syntheses from 13% to 70% of total area. According to the AFM data, the measured thickness for the layer of PVPyr was 6.0 nm; the layer consists of small globular structures (Figure 1C). Thickness was measured by scratching the soft polymer layer, measuring data, flattening the AFM image’s background, and measuring the peaks’ centers on the height distribution function for the scratch and its neighborhood region. The thickness of the layer with Au nanostars aggregates varies with the aggregate size and can be up to 0.78 μm in the highest points of the surface (Figure 1D). The suspension of freshly synthesized gold nanostars had a broad absorption spectrum with a maximum at approximately 900 nm (Figure 1E). Deposition of gold nanostars onto the polymer layer resulted in a 60–80 nm shift of peak maximum to the left. Au nanostar aggregates were less stable without the polymer layer and could be partially washed away by distilled water or cell culture medium.

The obtained substrates were tested for SERS performance by measuring the signal from 4-mercaptobenzoic acid at a 10${}^{-5}$ M concentration. The result of such testing is shown in the Appendix A. A slight increase in the signal from the analyte in the regions of large aggregates of gold nanostars on the surface of the substrate was observed.

### 3.2. Spectral Analysis

HeLa cells were seeded onto SERS substrate and incubated overnight to ensure good cell adhesion to the surface. Before measurements, cells were imaged with the inverted microscope. No contamination was noticed in all dishes, which indicated the adequacy of sterilization procedure of cellular substrates with UV. The cells were well attached to the SERS substrate without changing their typical morphologies.

Nanostars’ aggregates are seen as big black spots on brightfield images, whereas other areas of SERS substrates were transparent. It is challenging to obtain Raman spectra from wells on substrates placed in a plastic Petri dish due to the intense peaks from the plastic at the bottom of the dish. A small silicon wafer was underlaid beneath the SERS sensor to block Raman signal from plastic. Cells on SERS substrate placed on dark silicon were seen as shadows through the air-water interface under the Raman microspectrometer. Despite the cell shape being visible, it was challenging to recognize particular structures in a cell. A typical HeLa cell is about 20 μm diameter with a 10 μm nucleus located in the center of a cell. Therefore, spectra collected from the center of a cell were considered to originate mainly from the nucleus, whereas spectra collected from regions surrounding the nucleus corresponded to the cytoplasm. It can be assumed that due to differences in Au nanostar tip length and sharpness, some tips could penetrate a cell wall, some remained in the plasma membrane, and some did not contact the cell. Pylaev et al. [37] have already shown that Au nanostars can penetrate the cell wall and deliver vectors coding GFP proteins inside the cell. In our experiments, we used Au nanostars synthesized according to a similar protocol; thus, we assume that the mechanism of penetration into the cell wall remains the same.

The most intensive SERS spectra were obtained from cells adsorbed on large aggregates of Au nanostars. No peaks from cells placed on a modified or unmodified coverslip were detectable at the same detection parameters without Au nanostructures. The typical spectrum of a cell without aggregates of nanostructures is shown in the Appendix A.

It should be noted that there was a signal only from regions of a cell placed on aggregates of gold nanostars on PVPyr. Short-distance enhancement (1–2 nm) is supposed to be induced by the direct interaction of nanostars’ tips with molecules inside cells [35]. This can be achieved by gold nanostars’ tips penetrating the cell through the plasma membrane and cellular organelles’ membranes. Although the membrane is about 10 nm thick, penetration is possible because the average tip length is about 37 nm, and some reach lengths of up to 50 nm, according to the SEM image (Figure 1B).

The number of tips and, consequently, the probability of their penetration into the cell, was greater in aggregates of nanostructures than in individual nanoparticles. The rough morphology of aggregates contributed to plasma membrane stretch, which facilitated the penetration of tips. However, neither before nor after SERS imaging was cytoplasmic swelling or leakage observed. The small width of the tips allows them to penetrate the membrane without forming pores in it.

A 785 nm laser focused through the 50×/0.5 n.a. objective has been used for Raman measurements with a microspectrometer that resulted in an Airy disk radius of 1.4–1.9 μm. Since the diameter of the detection spot exceeds 1.4 μm, and mitochondria are located close to the nucleus, the appearance of peaks from mitochondria, RNA, and DNA in one spectrum may indicate that some of the Au nanostars in a registration spot enhance the signal from the nucleus, and some enhance the signal from surrounding mitochondria, which makes it possible to study them simultaneously.

Due to the significant decay of the SERS signal with distance from the nanostructure’s surface [44], we suggest that z-resolution depends only on the morphology of aggregates and distance of enhancement, despite the laser wavelength of 785 nm having good penetration ability.

SERS spectra from different parts of a cell on the aggregate of nanostructures were significantly different. Such a result was expected due to differences in molecular composition in a particular region of a cell. For example, single spectra from 4 points of the same cell are shown in Figure 2. Each Raman peak corresponds to a specific atom group vibration in molecules. Detailed peak assignments according to literature data are summarized in Appendix A [13,45,46,47,48,49,50,51,52,53,54].

Taking into account peak assignment and peak intensities of pure chemicals [46,50,51,52], and other compounds (Appendix A), we propose to consider spectral regions around 660–690, 790–805, and 1300–1350 cm${}^{-1}$ as originating primarily from bond vibrations in DNA nucleotides; 805–850 and 1510 cm${}^{-1}$ from RNA nucleotides; 730–765 cm${}^{-1}$ from hemes in mitochondrial cytochromes; 700–730, 1300–1330, and 1430–1470 cm${}^{-1}$ from lipids; 1000–1010 and 1360–1370 cm${}^{-1}$ from proteins.

Spectrum collected from the nucleus in the center of a cell contains intensive peaks of nucleotides (Figure 2G, spectrum 3). Peak 720 cm${}^{-1}$ may be attributed to adenine in DNA and phospholipids at the same time [51]. However, we did not observe the most intensive lipid peak 1445 cm${}^{-1}$, so peak 720 cm${}^{-1}$ can be considered characteristic of adenine. Peak 790 cm${}^{-1}$ is the most intensive peak of cytosine in DNA. Peak 645 cm${}^{-1}$ related to [Fe–S] cluster in mitochondrial electron transfer chain [47] is highly intensive. Peaks of hemes in mitochondrial cytochromes 745–760 and 1206 cm${}^{-1}$ are also presented in spectrum 1. According to Brazhe et al. [51], peak 750 cm${}^{-1}$ is more specific to heme C, whereas peak 760 cm${}^{-1}$ to heme B in mitochondria. The presence of specific peaks both from DNA and mitochondria opens a possibility to perform simultaneous analysis of mitochondria metabolism and local DNA composition. Intensive peaks 812 and 1510 cm${}^{-1}$ are most likely specific to RNA [48,51]. Spectrum 1 presumably represents cytoplasm composition. Spectrum 4 contains intensive peaks at 1141, 1225, 1350 cm${}^{-1}$ originating mainly from proteins and lipids [13,51,52,54]. Peaks 775 and 833 cm${}^{-1}$ from RNA are also presented [51,55]. Thus, this spectrum is supposed to represent the plasma membrane and a region of submembrane cytoplasm.

The regions surrounding the aggregate of nanostructures were chosen for SERS imaging. After baseline subtraction, the root-mean-square (RMS) value was calculated for Raman intensities in specific frequency ranges: 655–680 cm${}^{-1}$ corresponding to bond vibrations in DNA molecules; 740–765 cm${}^{-1}$ (hemes in mitochondria); 825–835 cm${}^{-1}$ (RNA), and 995–1010 cm${}^{-1}$ (proteins). Since there are no peaks in the spectral region 500–520 cm${}^{-1}$, this region may be considered noise. RMS values were normalized by the RMS value calculated for Raman intensities in the 500–520 cm${}^{-1}$ frequency range. Images of cells with these normalized RMS values can be used to locate DNA, mitochondria, RNA, and proteins, respectively, inside a cell (Figure 2A–D). To present the distribution of RNA, DNA, and mitochondria within an area of image registration, merged RGB images were made by overlaying images corresponding to DNA (red), RNA (green), mitochondrial hemes (blue) in the same range of RMS values (Figure 2E).

Aggregates of nanostructures varied in size and shape. They were located in the center of a cell (Figure 2F and Figure 3F) or on the border of a cell (Figure 3C). Figure 3 illustrates a significant variation of spectra depending on a region in a cell from which they were collected. Spectrum 1 was registered from nanostructures outside a cell. It contains small peaks at 1010, 1065, and 1203 cm${}^{-1}$, which have a negligible impact on the overall SERS spectra of cells.

Spectrum 1 was registered from nanostructures outside a cell. It contains small peaks at 1010, 1065, and 1203 cm${}^{-1}$, which have a negligible impact on the overall SERS spectra of cells. Spectra 2 and 3 look similar. Spectrum 3 contains a very intensive peak at 1446 cm${}^{-1}$ assigned to the C−H_2_ bend mostly in lipids and proteins; peak 1270 cm${}^{-1}$ corresponds to unsaturated fatty acids [50], and 1078 cm${}^{-1}$ corresponds to alkyl C−C gauche stretches in lipids [52]. Additionally, this spectrum contains intensive peaks from proteins: 1180, 1200, and 1570 cm${}^{-1}$ [48]. The carbohydrate peak at 1120 cm${}^{-1}$ is considered to originate from glycoproteins [52]. This may indicate that the spectrum represents the plasma membrane composition. Spectrum 2 contains less intensive peaks from lipids and another carbohydrate peak at 912 cm${}^{-1}$ [52], so it probably corresponds to another region of the plasma membrane with a more significant number of glycoproteins. Spectrum 4 contains peaks 1146 and 1495 cm${}^{-1}$, corresponding to C−C and C−N vibrations, respectively, inherent to different molecules [13,49,54]. Peak 833 cm${}^{-1}$ most likely corresponds to bond vibrations in RNA. Thus, spectrum 4 represents the composition of cytoplasm near the edge of the cell.

Spectra 5–7 are from the central part of the cell (Figure 3F). Peaks corresponding to RNA (822 cm${}^{-1}$) and DNA (670, 692, 802, 1330 cm${}^{-1}$) nucleotides are presented in all spectra of this region, indicating penetration of Au nanostars into the nucleus. Apart from them, there are intensive peaks of proteins (1000, 1180, and 1360–1370 cm${}^{-1}$) and some peaks of lipids (1260–1280, 1445 cm${}^{-1}$) presumably from the nuclear membrane. Peaks from hemes (746 and 754 cm${}^{-1}$) can be noticed in spectra 6 and 7.

In order to make sure that we are recording spectra in the cytosol of the cell, the cells of the murine melanoma B16-F10 were measured on SERS substrates. These cells contain melanin grains in the cytosol, and the presence of a characteristic enhanced SERS spectrum indicates that the SERS signal is indeed recorded from the intracellular contents. The corresponding analysis is shown in Appendix A.

### 3.3. Image Analysis

Normalized RMS values in frequency ranges specific to certain molecules in each pixel of Raman imaging were used to locate intracellular DNA, RNA, proteins, and mitochondrial hemes. Regions containing nuclei provided more intensive spectra compared to cytoplasm. If an aggregate was located near the edge of a cell, then there were less intensive spectra. Such an effect can be observed when comparing image (Figure 3A) with images (Figure 2D and Figure 3D) where protein distribution is visualized with normalized RMS values in the frequency range 995–1010 cm${}^{-1}$. As peripheral parts of a cell are more flexible, less dense, and heavier than the central part, the penetration of tips there may be less profound.

There was no aggregate in the center of a cell (Figure 3C), unlike in the following cases. The maximal RMS value for proteins was achieved in Figure 3D, whereas the minimum value was observed in the case of image acquisition outside the nucleus (Figure 3A).

RMS values of frequency ranges related to vibrations in DNA and RNA molecules were the highest within a nucleus and surrounding cytoplasm (Figure 2A,C,E). In Figure 3E, the only intensive region is associated with the nucleus. Areas with intensive RNA peaks (green) in merged images may be interpreted as cytoplasm or endoplasmic reticulum containing many ribosome-synthesizing proteins. This region may also be suggested as the nucleolus, the largest structure in the nucleus, containing a large amount of ribosomal RNA. Figure 3E potentially demonstrates this.

The signal from heme vibrations in mitochondrial cytochromes was weaker than from other molecules. However, that does not mean that there were few mitochondria in cells. Wavelength 785 nm is not resonant for hemes; therefore, peaks at the 740–765 cm${}^{-1}$ range are weak. Spots related to mitochondria (hemes) were located mainly outside nuclei since mitochondria usually surrounded them.

### 3.4. Nanostars Localization Inside Cells

To better see the localization of aggregates inside HeLa cells, 3D images were taken using confocal laser scanning microscopy (Figure 4). The cytosol of the cells was stained with the vital dye calcein AM, which provides uniform staining of the internal contents of the cells. Gold nanoparticles and their aggregates are visible against the green background as dark silhouettes. It should be noted that large aggregates are sufficiently opaque for optical radiation and the obscure part of the cell behind them, which makes it difficult to localize particles along the Z axis.

To sum up, SERS spectra from different cell parts originated from the molecular environment around gold tips that penetrated the cell. The number of tips and the depth of their penetration were significantly higher on the surface of aggregates. The SERS signal, even from DNA, RNA, and hemes located far from the plasma membrane, can be detected using nanostars with long tips up to 50 nm.

## 4. Conclusions

We have demonstrated a novel planar SERS substrate based on gold nanostar aggregates immobilized in the poly(4-vinyl pyridine) layer on the glass substrate. Tips of Au nanostars penetrate the cell membrane and provide the SERS signal from biomolecules in living HeLa cells. This platform provides an excellent alternative to other methods of SERS sensor internalization and can be used to investigate the biochemistry of different cellular organelles, including the nucleus with the nucleolus and mitochondria. The proposed nanostructured surfaces provide information about the distribution of different molecules in cellular compartments without any labeling.

## Figures and Tables

**Figure 1 nanomaterials-11-02588-f001:**
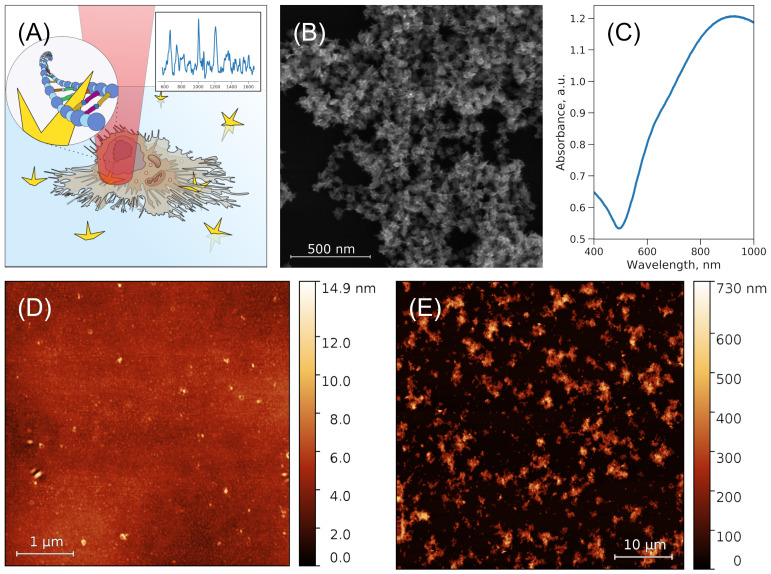
(**A**) Experimental scheme: a cell is attached to a nanostructured surface consisting of gold nanostars deposited on a PVPyr layer on a coverslip. (**B**) SEM image of nanostructured surface. (**C**) Absorption spectra of a freshly synthesized suspension of gold nanostars. (**D**) AFM image of the PVPyr layer and (**E**) PVPyr layer with Au nanostars.

**Figure 2 nanomaterials-11-02588-f002:**
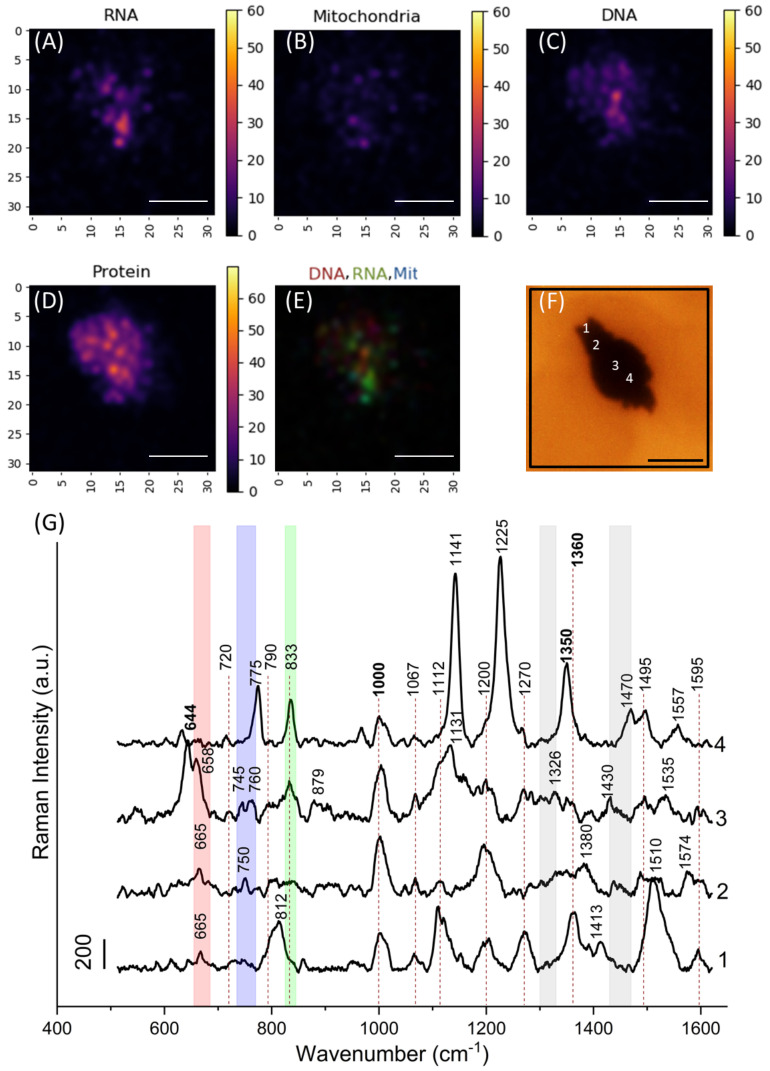
(**A**–**D**) Raman images of the chosen region (a black rectangle on a microphotograph of a cell on nanostructures (**F**)). Each point of the image corresponds to the normalized root-mean-square (RMS) value calculated for frequency ranges: (**A**) 825–835 cm${}^{-1}$ (RNA); (**B**) 740–765 cm${}^{-1}$ (mitochondria); (**C**) 655–680 cm${}^{-1}$ (DNA); and (**D**) 995–1010 cm${}^{-1}$ (proteins). (**E**) The merged RGB image was made by overlaying grayscale images corresponding to RNA (green), mitochondria (blue), and DNA (red) in the same range of RMS values (0–60). Axes show a pixel number. The scale bars at (**A**–**F**) correspond to 10 μm. (**G**) SERS spectra from regions of interest (numbered in (**F**)). Rectangles are related to certain frequency ranges: 825–835 cm${}^{-1}$ (green), 740–765 cm${}^{-1}$ (blue), 655–680 cm${}^{-1}$ (red), 1300–1330 cm${}^{-1}$ and 1430–1470 cm${}^{-1}$ (gray), which are highly specific to bond vibrations in RNA, mitochondria, DNA and lipids, respectively. Peaks from proteins are shown in bold.

**Figure 3 nanomaterials-11-02588-f003:**
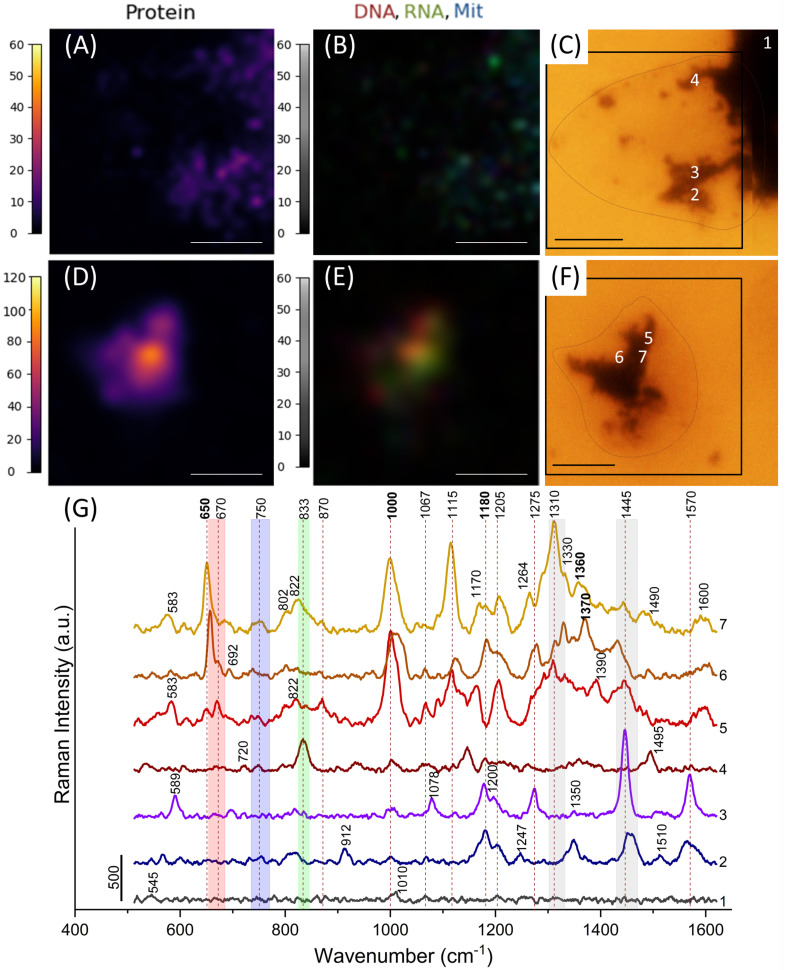
(**A**,**D**) Normalized RMS calculated for the frequency range 995–1010 cm${}^{-1}$ suggested as protein distribution for regions of image acquisition (**C**) and (**F**), respectively. (**B**,**E**) Overlapped normalized RMS images of DNA (red), RNA (green), and mitochondria (blue) in the same scale 0–60 for regions (**C**) and (**F**), respectively. (**C**,**F**) Microphotographs of regions of image acquisition. The scale bars at (**A**–**F**) correspond to 10 μm. Edges of cells are circled with a dashed line. (**G**) Spectra of points of interest (numbered on microphotographs (**C**) and (**F**)) extracted from SERS images. Rectangles designate peaks from atom group vibrations in DNA (red), RNA (green), mitochondria (blue), lipids (gray). Peaks from proteins are shown in bold.

**Figure 4 nanomaterials-11-02588-f004:**
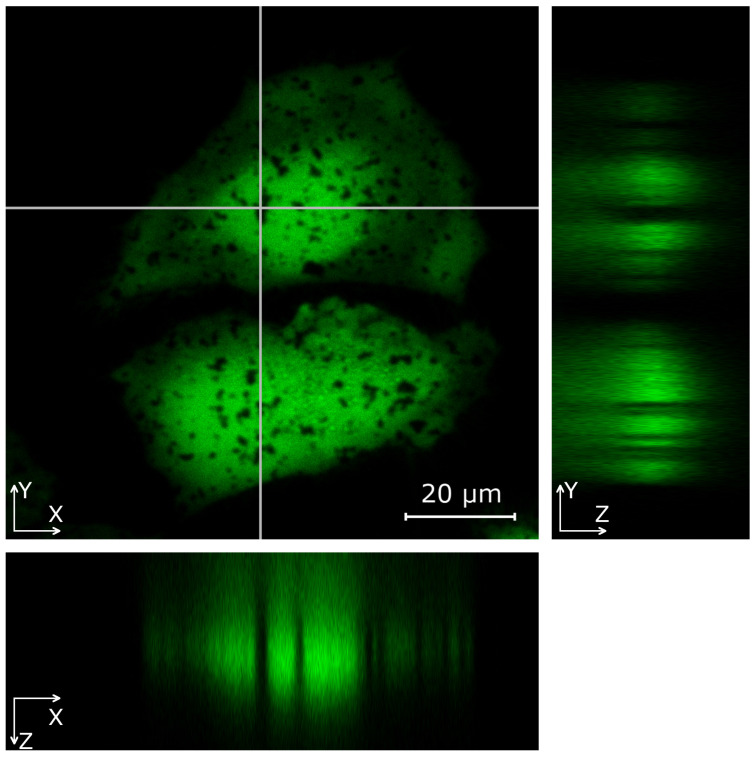
CLSM 3-D image of HeLa cells on the substrate with gold nanostars aggregates.

## Data Availability

Data underlying the results presented in this paper are not publicly available at this time but may be obtained from the authors upon reasonable request.

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
