# Peer review of "Live Cell Poration by Au Nanostars to Probe Intracellular Molecular Composition with SERS"

_nanomaterials, 2021, doi:10.3390/nano11102588_

Round 1
Reviewer 1 Report
This is an important study and development of a new nanomaterial which allows for SERS-based probing of intracellular molecules. The Au nanostars can penetrate the cell membrane and generate SERS signals inside living cells. This is a novel tool for cell research that provides Raman probing of proteins, lipids, DNA, RNA, and other molecules inside cells and different cell compartments. The data set presented in the manuscript is impressive. The manuscript is well written and may be published as is.
Author Response
Answer to the Reviewer's Comments
This is an important study and development of a new nanomaterial which allows for SERS-based probing of intracellular molecules. The Au nanostars can penetrate the cell membrane and generate SERS signals inside living cells. This is a novel tool for cell research that provides Raman probing of proteins, lipids, DNA, RNA, and other molecules inside cells and different cell compartments. The data set presented in the manuscript is impressive. The manuscript is well written and may be published as is.
We are grateful for the Reviewer's comment. We truly appreciate this opinion.
Reviewer 2 Report
The paper reports on SERS analysis of living cells for the identification of intracellular components. The paper could be of interest but i requires major revisions. Some procedures must be better described in Materials and Methods section. How have AuNS been deposited on the polymer surface? Please, report the procedure and comment on the reproducibility of the samples since the processes not controlled.
The analysis of the results should be improved since it is not very convincing. The attribution of Raman peaks in the registered spectra is not so straightforward as presented by the authors and the references to other published materials not really enough. Authors should present the spectra of single components (DNA, RNA, proteins) isolated from cells and then compare the spectra acquired in the living ones. The SERS substrate should be also tested with some standard molecules in order to measure the enhancement factor and how the molecules are arranged on it
The Raman images should have the length bar since it is not clear how big is the area reported in Fig 2A-E and Fig.3 A-B, D-E.
It is not clear how authors could distinguish the presence of the Au NS in different positions of the cells. What did authors mean by considering similar the spectra 2 and 3 (they have different peaks at 1200 and 1380 cm-1, for example)? Please, comment better this point.
The proposed SERS substrate could be of interest but it requires a better characterization and a deeper study on the spectra interpretation of the living cells.
Author Response
Some procedures must be better described in Materials and Methods section.
We have changed Materials and methods section and added some extra materials to the article.
How have AuNS been deposited on the polymer surface? Please, report the procedure and comment on the reproducibility of the samples since the processes not controlled.
We have provided additional information about nanostars deposition, its characterization with standard test molecules for SERS on gold nanoparticles. Also we have added some reproducibility comments to the article.
The analysis of the results should be improved since it is not very convincing. The attribution of Raman peaks in the registered spectra is not so straightforward as presented by the authors and the references to other published materials not really enough. Authors should present the spectra of single components (DNA, RNA, proteins) isolated from cells and then compare the spectra acquired in the living ones.
The isolation and purification of single molecules from this type of cells seems too complex task to do in a week time given us to reply the reviewer comments. So we decided just to add another control experiment with B16-F10 murrine melanoma cells containing melanin molecules inside the cell in form of the melanin grains, so the clear SERS spectra of melanin can be obtained only if enhancement is done inside the cell volume, not on the membrane. It seems to be a good clarification that we obtain the clear spectra of cell internal molecules with this type of substrate.
The SERS substrate should be also tested with some standard molecules in order to measure the enhancement factor and how the molecules are arranged on it
We have added characterisation of SERS substrate with the standard test based on 4-MBA (4-Mercaptobenzoic acid) molecules.
The Raman images should have the length bar since it is not clear how big is the area reported in Fig 2A-E and Fig.3 A-B, D-E.
We have added the lateral scale bars to this images.
It is not clear how authors could distinguish the presence of the Au NS in different positions of the cells.
We have stained the cell internal volume with the calcein AM and took the CLSM images to make it more clear, the stars and its clusters become visible as dark shades of gold shapes and fluorescence quenching around it. Additional Fig. 4 is provided in the article.
What did authors mean by considering similar the spectra 2 and 3 (they have different peaks at 1200 and 1380 cm-1, for example)? Please, comment better this point.
We have just removed this claim, it seems misleading in the text.
Reviewer 3 Report
Authors investigated intracellular molecular composition inside living cells by surface-enhanced Raman scattering (SERS) of gold nanostars.
The SERS substrate is formed by stabilized aggregates of gold nanostars on a glass microscope slide coated with a layer of poly (4-vinyl pyridine) polymer.
However authors should describe the following:
Authors used aggregates of gold nanostars for surface-enhanced Raman scattering platform. Usually aggregates of gold nanostars exhibit poor repeatability and inaccuracy. Authors should describe repeatability and accuracy of SERS-based sensing by aggregates of gold nanostars.
It is required to compare SERS-based sensing by aggregates of gold nanostars with SERS-based sensing by gold nanoparticles.
Authors should describe amount of gold nanostars inside biological components (DNA, RNA, Protein, Mitochondria) of Hela cells.
Author Response
Authors used aggregates of gold nanostars for surface-enhanced Raman scattering platform. Usually aggregates of gold nanostars exhibit poor repeatability and inaccuracy. Authors should describe repeatability and accuracy of SERS-based sensing by aggregates of gold nanostars.
It is required to compare SERS-based sensing by aggregates of gold nanostars with SERS-based sensing by gold nanoparticles.
We have provided the standard test of this platform with 4-mercaptobenzoic acid (4-MBA, 10^-5 M) molecules in the new supplementary materials part. The obtained platform is definitely less uniform and stable comparing to the uniformly distributed nanoparticles or for example gold leaflets overgrown to very small gaps between them. However, the main new properties arising from using the particle agregates is ability to penetrate inside the volume of the cell and measure its internal volume. it seems to authors that this new property provide substantial new functionality to the platform, in price of making it less uniform and regular.
Authors should describe amount of gold nanostars inside biological components (DNA, RNA, Protein, Mitochondria) of Hela cells.
We have added new part with CLSM imaging of cells stained with Calcein-AM making gold aggregates inside the different cell compartments clearly visible against the uniform Calcein background. Shading by non-transparent large aggregates make their Z-position still a bit controversal, but we think we have provided enough evidence that we're measuring signals from the cell internal volume.
Round 2
Reviewer 2 Report
The revised paper can be published
Author Response
The revised paper can be published
We are grateful for this opinion.
Reviewer 3 Report
Authors revised most of manuscript according reviewer's comments.
Author's response includes the following:
"However, the main new properties arising from using the particle agregates is ability to penetrate inside the volume of the cell and measure its internal volume. it seems to authors that this new property provide substantial new functionality to the platform, in price of making it less uniform and regular."
It seems that average size of gold nanostar aggregates is larger than those of gold nanoparticles, and gold nanostar aggregates have complex structure than spherical gold nanoparticles. Therefore, it is difficult for gold nanostar aggregates to penetrate inside the volume of the cell and measure its internal volume. Authors should add experimental results or relevamt references to support the assumption that the particle agregates can penetrate inside the volume of the cell and measure its internal volume.
Author Response
It seems that average size of gold nanostar aggregates is larger than those of gold nanoparticles, and gold nanostar aggregates have complex structure than spherical gold nanoparticles. Therefore, it is difficult for gold nanostar aggregates to penetrate inside the volume of the cell and measure its internal volume. Authors should add experimental results or relevamt references to support the assumption that the particle agregates can penetrate inside the volume of the cell and measure its internal volume.
Indeed, gold nanostars aggregates are larger than individual nanoparticles (spherical ones as well as individual nanostars). At the same time, the surface of gold nanostars aggregates at the nanoscale level is similar to single nanostar according to both scanning electron microscopy and atomic force microscopy. The fact that gold nanostars perforate the cell membrane is shown in the article by Tim Pylaev et al. (ref. 37), where a vector encoding a GFP was introduced along with the stars. The mechanism of membrane penetration remains controversial, but the most feasible hypothesis is the local thermal effect of the tips of the nanostars, in which the plasmon resonance is excited. We similarly show this with cells B16-F10 (Fig. S3).